



# On the determination of ionospheric electron density profiles using multi-frequency riometry

Derek McKay[1,2], Juha Vierinen[3], Antti Kero[4], and Noora Partamies[5,6]

[1]FINCA, Turku University, Turku, Finland
[2]Metsähovi Radio Observatory, Aalto University, Kylmälä, Finland
[3]Department of Physics and Technology, University of Tromsø, Norway
[4]Sodankylä Geophysical Observatory, University of Oulu, Finland
[5]UNIS University Centre in Svalbard, Svalbard, Norway
[6]Birkeland Centre for Space Science, Bergen, Norway

**Correspondence:** D. McKay (derek.mckay@utu.fi)

**Abstract.** Radio wave absorption in the ionosphere is a function of electron density, collision frequency, radio wave polarisation, magnetic field and radio wave frequency. Several studies have used multi-frequency measurements of cosmic radio noise absorption to determine electron density profiles. Using the framework of statistical inverse problems, we investigated if an electron density altitude profile can be determined by using multi-frequency, dual-polarisation measurements. It was found that

the altitude profile cannot be uniquely determined from a "complete" measurement of radio wave absorption for all frequencies and two polarisation modes. This implies that accurate electron density profile measurements cannot be ascertained using multi-frequency riometer data alone, but that the reconstruction requires a strong additional a priori assumption of the electron density profile, such as a parameterised model for the ionisation source. Nevertheless, the spectral index of the absorption could be used to determine if there is a significant component of hard precipitation that ionises the lower part of the D region, but it

is not possible to infer the altitude distribution uniquely with this technique alone.

## 1 Introduction

Jansky (1933) determined that certain detected radio noise was of cosmic origin, and was associated with the Galaxy, thus founding a new branch of science — Radio Astronomy. Shain (1951) observed the absorption of this cosmic radio noise by

the ionosphere. This led to the development of new instruments specifically designed for using this phenomenon to investigate the ionosphere. The first dedicated instruments to measure the absorption effect were developed shortly thereafter (e.g. Machin et al., 1952). One later example built by Little and Leinbach (1959) — the R.I.O.M.E.T.E.R (for *Relative Ionospheric Opacity Meter for Extra-Terrestrial Emissions of Radio noise*) — gave the name "riometer" to this generic class of instrument and the term "riometry" for the measurement of cosmic noise absorption (CNA) by the ionosphere.





The riometer is a stable radio receiver with a known beam-pattern. It operates at some frequency just above the radio penetration frequency of the atmosphere so that it can detect cosmic radio noise. Reductions in received power are a result of the signal being absorbed by the ionosphere. Anomalous absorption is determined by comparing the received signal to the signal that would be expected as the result of transmission through an undisturbed ("quiet") ionosphere. At times of ionospheric disturbance, such as during aurorae or other particle precipitation events, increases in the electron density cause
enhanced absorption of the radio signals (e.g. Hunsucker, 1991).

The approach of using riometer measurements at multiple frequencies to determine the electron density profile was first proposed by Parthasarathy et al. (1963). In addition to observations made with discrete frequencies, it is possible to make spectral absorption measurements. Belikovich et al. (1964) demonstrated that a very broad frequency range was required to determine the electron density profile for 40–80 km, such that it was effectively impossible using this single method and that pulse-
sounding measurements are needed to supplement the measurement. Absorption heights determined from multi-frequency riometry are therefore lower limits and not full profiles (Hargreaves, 1969). Measurements using a continuous spectrum of radio frequencies (spectral riometry) has been compared to an alternate method (incoherent scatter radar) to validate its measurement of electron density enhancement (Kero et al., 2014). In that study, the observed absorption spectrum was used to invert the corresponding electron density profile by applying a simple parameterised electron precipitation model. The comparison
with the nearby incoherent scatter radar indicated that multi-frequency riometry could determine comparable electron density profiles based on the posteriori probability distribution of two free parameters (electron precipitation energy and flux), based on the least-squares fit between the measured absorption spectrum and the parameterised model. Kero et al. (2014) concluded that the spectral riometry approach is capable of producing realistic electron density profiles under conditions of substorm-related electron precipitation, but noted a correlation between the two parameters, pointing at the fact that different precipitation pa-
rameter pairs can produce approximately equal electron density profiles at the altitudes of the maximum absorption, suggesting potential mathematical degeneracy. Since then Martin et al. (2016) used a Bayesian method and obtained good agreement with incoherent scatter results. However, it was noted that the determined profiles were similar to the simulated profiles.

In this study, we utilise the framework of statistical inverse problems (Kaipio and Somersalo, 2006) to study how well the electron density profile of the lower ionosphere can be determined using a multi-frequency riometer. The measurement will
first be formulated as a linear inverse problem. The a posteriori error covariance matrix will then be investigated to determine what is the distribution of errors when estimating the electron density profile.

## 2   Radio absorption

The Appleton-Hartree equation provides a formula for the refractive index for radio waves propagating in a collisional plasma (Hunsucker and Hargreaves, 2002). The absorption is mainly due to electron-neutral collisions in the D region of the ionosphere. In
the F region, electron-ion collisions dominate. The absorption $A$, in decibels (dB), is given by:

$$A = 10\log_{10}(e)\left(\frac{q_e^2}{m_e\epsilon_0 c}\right)\int_L \left(\frac{N_e\nu_{en}}{\nu^2 + (\omega \pm \omega_L)^2}\right)\left(1 - \frac{\omega_p^2}{\omega^2}\right)^{-\frac{1}{2}} d\boldsymbol{l}, \tag{1}$$





where $e \approx 2.72$, $q_e$ is the charge of an electron, $m_e$ is the mass of an electron, $\epsilon_0$ is the permittivity of free space, $c$ is the speed of light, $L$ is the path of the radio wave, $d\boldsymbol{l}$ is an infinitesimal line element along $L$, $N_e$ is the electron density, $\nu$ is the sum of effective electron-neutral ($\nu_{en}$) and electron-ion ($\nu_{ei}$) collision frequencies, $\omega$ ($= 2\pi f$) is the radio wave angular frequency, and $\omega_p = \sqrt{n_e q_e^2 / m_e \epsilon_0}$ is the plasma frequency. The term $\omega_L$ is the component of the electron gyro-frequency parallel to the magnetic field, from $\omega_H \cos\theta = \omega_L$, where $\omega_H$ is the gyro-frequency and $\theta$ is the angle between the magnetic field and the direction of propagation (Hargreaves, 1969).

Before reaching the Earth, the cosmic radio noise is unpolarised. However, on passing through the ionosphere, the extraordinary mode (x-mode) signal will incur more absorption than then ordinary mode (o-mode) signal (Little et al., 1964). This is manifested in Eqn. 1 as the $\pm\omega_L$ term, with $+\omega_L$ corresponding to o-mode and $-\omega_L$ corresponding to x-mode.

In all practical riometer observations, the following term is nearly unity:

$$\left(1 - \frac{\omega_p^2}{\omega^2}\right)^{-1/2} \approx 1. \tag{2}$$

This is assumed in this study as well. Additionally, absorption occurs at the lower altitudes, where electron-neutral collisions dominate, and thus the formula for absorption simplifies to:

$$A = 4.611 \times 10^{-5} \int\limits_L \frac{N_e \nu_{en}}{\nu_{en}^2 + (\omega \pm \omega_L)^2} d\boldsymbol{l}. \tag{3}$$

This equation is widely used when modelling CNA (Hargreaves, 1969; Hunsucker and Hargreaves, 2002). The electron collision frequency is ultimately a function of neutral density and electron temperature. However, at any given altitude, the electron temperature typically changes very little, and variations in absorption are attributed to variations in electron density. A measure of the electron collision frequency, $\nu$, is required. For this the results collated by Aggarwal et al. (1979, Fig.7) were used and a look-up table was generated and then linear interpolation was used for determining intermediate values. This gives a realistic electron collision frequency for any given height in the approximate range of 50–500 km and is shown in Figure 1.

## 2.1 Specification of the forward model

The forward problem is defined as:

$$\boldsymbol{d} = \boldsymbol{G}\boldsymbol{m} + \eta \tag{4}$$

where $\boldsymbol{d}$ is the data, $\boldsymbol{G}$ is the forward model and $\boldsymbol{m}$ is the model. For generating sample data, an error term, $\eta$, is used, which follows a normal distribution. In this study, the data, $\boldsymbol{d}$, is the absorption measured at a given frequency and polarisation mode. The model, $\boldsymbol{m}$, is the electron density, $N_e(h)$ for a given height $h$. The range of heights used can be varied, but for this initial test a range of 65–110 km was selected, which spans the D region, and includes lower altitudes down to 50 km (which would be subject to electron-density enhancements, in the event of hard precipitation).





**Figure 1.** The model of electron collision frequency $\nu = \nu_{en} + \nu_{ei}$ from Aggarwal et al. (1979).

80    The forward model, $\boldsymbol{G}$, is a linear algebraic representation of the riometry equation (Equation 3). The riometry equation is continuous, so it is discretised as:

$$A = 4.611 \times 10^{-5} \sum_{h=h_{\min}}^{h_{\max}} \frac{N_\text{e}\nu}{\nu^2 + (\omega \pm \omega_L)^2}\, \Delta h\ \text{(dB)} \tag{5}$$



This can be expressed in matrix form as

$$
\begin{bmatrix}
A_{o_1} \\
A_{o_2} \\
\vdots \\
A_{o_N} \\
A_{x_1} \\
A_{x_2} \\
\vdots \\
A_{x_N}
\end{bmatrix}
=
\begin{bmatrix}
\frac{k\nu_1\Delta h}{\nu_1^2+(\omega_1+\omega_L)^2} & \frac{k\nu_2\Delta h}{\nu_2^2+(\omega_1+\omega_L)^2} & \cdots & \frac{k\nu_M\Delta h}{\nu_M^2+(\omega_1+\omega_L)^2} \\
\frac{k\nu_1\Delta h}{\nu_1^2+(\omega_2+\omega_L)^2} & \frac{k\nu_2\Delta h}{\nu_2^2+(\omega_2+\omega_L)^2} & \cdots & \frac{k\nu_M\Delta h}{\nu_M^2+(\omega_2+\omega_L)^2} \\
\vdots & \vdots & \ddots & \vdots \\
\frac{k\nu_1\Delta h}{\nu_1^2+(\omega_N+\omega_L)^2} & \frac{k\nu_2\Delta h}{\nu_2^2+(\omega_N+\omega_L)^2} & \cdots & \frac{k\nu_M\Delta h}{\nu_M^2+(\omega_N+\omega_L)^2} \\
\frac{k\nu_1\Delta h}{\nu_1^2+(\omega_1-\omega_L)^2} & \frac{k\nu_2\Delta h}{\nu_2^2+(\omega_1-\omega_L)^2} & \cdots & \frac{k\nu_M\Delta h}{\nu_M^2+(\omega_1-\omega_L)^2} \\
\frac{k\nu_1\Delta h}{\nu_1^2+(\omega_2-\omega_L)^2} & \frac{k\nu_2\Delta h}{\nu_2^2+(\omega_2-\omega_L)^2} & \cdots & \frac{k\nu_M\Delta h}{\nu_M^2+(\omega_2-\omega_L)^2} \\
\vdots & \vdots & \ddots & \vdots \\
\frac{k\nu_1\Delta h}{\nu_1^2+(\omega_N-\omega_L)^2} & \frac{k\nu_2\Delta h}{\nu_2^2+(\omega_N-\omega_L)^2} & \cdots & \frac{k\nu_M\Delta h}{\nu_M^2+(\omega_N-\omega_L)^2}
\end{bmatrix}
\cdot
\begin{bmatrix}
m_1 \\
m_2 \\
\vdots \\
m_M
\end{bmatrix}
\tag{6}
$$

In Equation 6, $m_1, m_2, \dots m_M$ are the model parameters; i.e. the electron density at a given height, $m_i = N_e(h)$, where $i$ is the height array index for a given height, $h$. The size of a height increment, $\Delta h$, is the distance between heights, $h_i$ and $h_{i+1}$. The data are the measured absorptions in decibels for the o- and x-mode polarisations; $A_{o_j}$ and $A_{x_j}$, respectively, where $j$ is the frequency channel array index. The series $\nu_1, \nu_2, \dots \nu_M$ is the electron collision frequency profile, where $\nu_i = \nu(h)$, where $i$ is the height array index for a given height, $h$. A look-up table is used for the collision frequencies (Figure 1). In order to de-clutter the representation, the symbol $k$ is used for the riometry constant from Equation 3, with $k = 4.611 \times 10^{-5}$. A diagramatic representation of the matrices and their dimensions is shown in Figure 2.

The angular gyrofrequency is a function of the magnetic field, which is effectively constant over the range of heights being studied. The magnetic field expected in the sub-polar regions of the Earth is typically $50\,\mu$T, giving an electron gyro frequency of $\omega_H = 2\pi \times 1.4 \times 10^6$ Hz. Ions and other larger species have much higher masses, resulting in gyro frequencies which can be neglected (McKay, 2018). Field-aligned observations are assumed (where $\omega_L = \omega_H$), as this is the best-case scenario.

## 2.2 Example electron density profiles

To test the model and the ability to solve it, synthetic data are generated for the altitude ranges that could be expected in a physical situation. The frequency range chosen, 15–78 MHz, extends from the penetration cut-off frequency to just below the FM radio band (a practical limit for riometers due to interference).

An electron density profile from Gnanalingam and Kane (1975) was used as a basis (the 'Normal' profile, shown in Figure 4). In order to simulate lower altitudes, a simple interpolation between the lowest value from Gnanalingam and Kane (1975) and that of Jespersen et al. (1964) was made. This is a safe assumption, as the electron densities at these altitudes are several orders of magnitude lower than those found in the D and E regions.

Although there is always some level of absorption imposed by the atmosphere on incoming radio waves, it is the enhanced absorption that is of particular interest and is also the target measurement for riometers. The electron density enhancements in the ionosphere are often the result of substorms (Hunsucker and Hargreaves, 2002). Earlier studies (e.g. Jussila et al., 2004) have determined that neither plasma instabilities nor enhanced electron temperatures in the E region play a significant role in





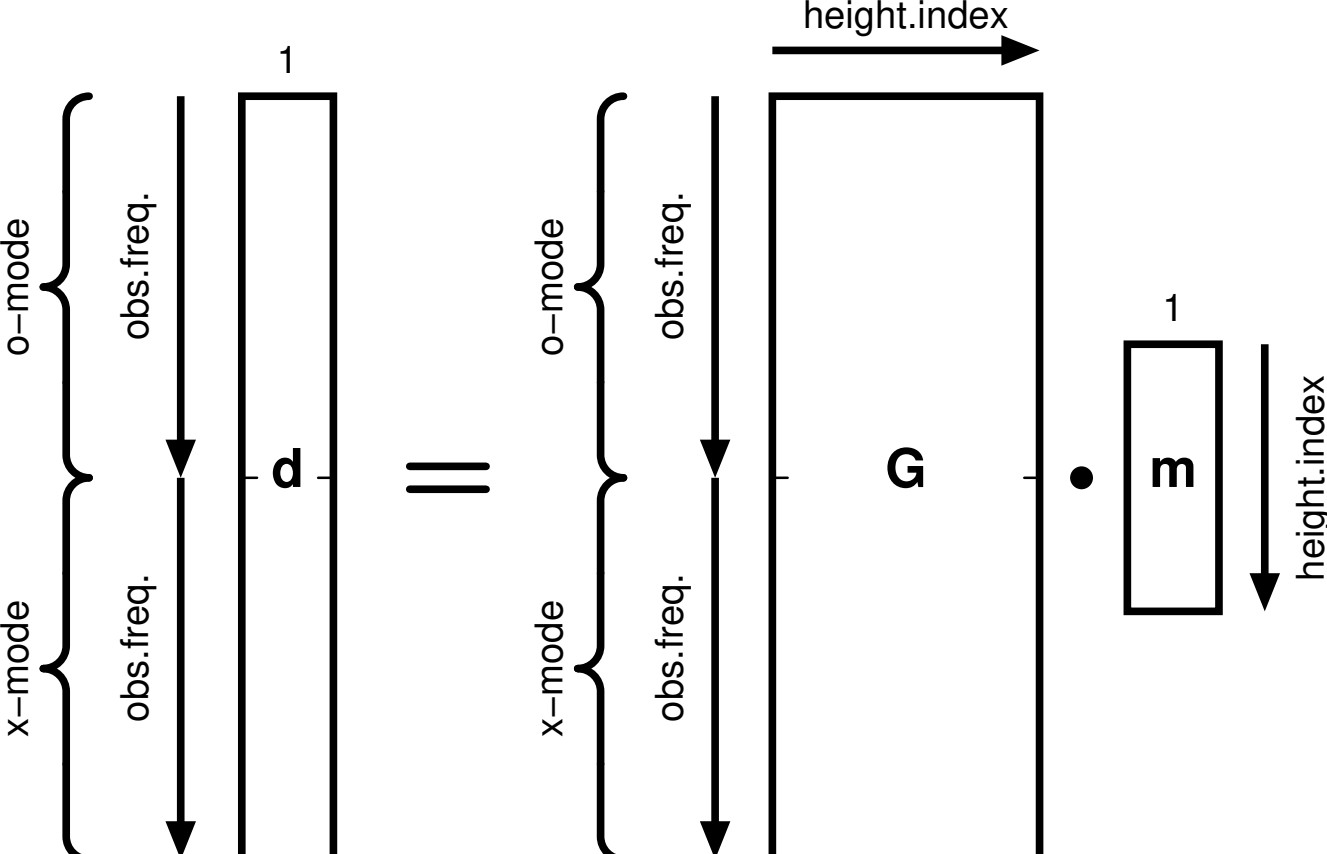

**Figure 2.** Illustration of the matrix dimensions of the model. (c.f. Equation 6)

causing the CNA and concluded that CNA is caused by energetic electron precipitation reaching down into the D region, with a maximum between 80 and 90 km.

An enhancement of approximately one order of magnitude is made at an altitude of 90 km, with a full-width-half-maximum of 10 km. This is applied to the "normal profile" based on Gnanalingam and Kane (1975) to simulate the "enhanced profile". These normal and enhanced profiles are shown in Figure 3.

### 2.3   Application of the forward model

With a model electron density profile it is now possible to apply the forward model, $G$, to it to generate the sample data. This
has been done for both the normal and enhanced profiles and the results are shown in Figure 4.

The first thing to note from these data is that there is already a background amount of absorption between the normal and enhanced electron density profiles. A riometric measurement, on the grounds that it uses a quiet-day subtraction technique, is measuring the difference in absorption between the normal and enhanced conditions.





**Figure 3.** Electron density profiles for "normal" and "enhanced" conditions.

The effect of absorption decreases with radio frequency, which is the reason why riometers operate in the 30–40 MHz range.

Above this, absorption is comparable to noise levels (±0.1 dB) from typical instruments.

The strongest absorption effects are at the lower frequencies, but when cross-referencing against the radio-frequency environment of suitable instruments (e.g. McKay-Bukowski et al., 2015, Fig.7), frequencies below 25 MHz are fraught with short-wave radio interference, making their use impractical.






**Figure 4.** Expected CNA from "normal" and "enhanced" conditions for both x-mode and o-mode radio waves. The horizontal axis the observing radio frequency, $\nu = \omega/2\pi$, in megahertz.

## 3    Inverse analysis

The method used to solve this inverse problem is singular value decomposition (SVD). Equation 4 is refactored as:

$$\boldsymbol{G} = \boldsymbol{U}\boldsymbol{S}\boldsymbol{V}^T \tag{7}$$

where $\boldsymbol{U}$ is an N-by-N orthogonal matrix with columns that are unit basis vectors in the data space, $\boldsymbol{V}$ is an M-by-M orthogonal matrix with columns that are basis vectors in the model space, and $\boldsymbol{S}$ is an N-by-M diagonal matrix; the diagonal elements are the singular values. The SVD is used to compute the Moore-Penrose pseudoinverse, such that:

$$\boldsymbol{G}^\dagger = \boldsymbol{V}_p \boldsymbol{S}_p^{-1} \boldsymbol{U}_p^T \tag{8}$$





where $\boldsymbol{V}_p$ represents the first $p$ columns of $\boldsymbol{V}$ (and similarly for the other matrices). This is a valid simplification, as the singular values of $\boldsymbol{S}$ are typically arranged in decreasing magnitude along the diagonal. By assuming that $\boldsymbol{S}$ can be represented as

$$\boldsymbol{S} = \begin{bmatrix} \boldsymbol{S}_p & 0 \\ 0 & 0 \end{bmatrix} \tag{9}$$

As columns $\geq p$ in $\boldsymbol{U}$ and $\boldsymbol{V}$ are multiplied by zeros in the $\boldsymbol{S}$ matrix, the matrices can be treated as orthonormal, with the simplifications that can be applied to orthonormal matrices. Thus the generalised form becomes:

$$\boldsymbol{G}^\dagger = \boldsymbol{V}_p \boldsymbol{S}_p^{-1} \boldsymbol{U}_p^T \tag{10}$$

The proof for which is given in Aster et al. (2011, Ch.3).

### 3.1 Interpretation of the SVD products

Following the singular value decomposition, the $\boldsymbol{V}$ matrix can be represented as an image to gain an understanding of the determinism of the solution. This is shown in Figure 5. There is a large noise component throughout most of the solution, with only the first few columns showing non-noise structure. The columns are the basis functions, with the first twenty of these being shown with scale information in Figure 6. As can be seen, the first few basis functions have structure, but thereafter the noise becomes increasingly dominant.

As Equation 5, and thus the forward matrix system represented by Equation 6, are linear systems, it is possible to assess them for rank deficiency — namely, insufficient information to extract the parameters of the desired model. Although higher order terms have non-zero values, these are extremely small and represent round-off errors and floating-point number quantisation errors.

In Figure 7, the eigenvalues of the $\boldsymbol{S}$ matrix are plotted as a function of the eigennumber to create a so-called "L-curve". As the values are so tiny and cover a large dynamic range, a log-linear plot is used to highlight the salient features, as a result distorting the original "L" form.

Typically, the technique is to truncate the effectively-zero singular values. This produces a least squares solution of limited resolution. Although rank deficient problems can be solved by applying a generalised inverse solution, there is insufficient information to recover the complexity of the model — in this application case: the electron density profile.

Examination of Figure 7 shows there are approximately two orders of magnitude difference between the first and second term. Even if there was information contained within those terms (and the basis functions suggest there might be to the fifth term) the effect that this has on the model determination is negligible.

Terms after the tenth term are below $10^{-24}$. Compared to the first term (just below $10^{-8}$), this represents a $10^{16}$ dynamic-range shift. This is equivalent to the numerical dynamic range of the double-precision floating-point number representation (bits $2 - 53 \approx 1.11 \times 10^{-16}$). Thus, these higher terms are equivalent to the least-significant bit of the floating-point representation and thus can be considered to be numerically zero.





**Figure 5.** Image representation of the **V** matrix. The first 11 columns show non-noise structure. These, with scale information, are shown in Figure 6.





**Figure 6.** The first 20 basis functions from the $V$ matrix.

As a result, it can be hypothesised that any number of model solutions could be formulated that would result in a superficial fit of the data. If this hypothesis holds, then it demonstrates the non-uniqueness of the solution, and thus will verify that the original inverse problem is ill-posed.

## 3.2 Inversion

To test the hypothesis, two test model data sets are formulated using literature-sourced data (Section 2.2) and a data set that can be derived by applying the forward model to those data (Section 2.3). These data sets include one for normal quiet ionospheric conditions, $d_{\mathrm{nor}}$ and one for enhanced conditions, $d_{\mathrm{enh}}$. The forward matrix, $G$, was applied and a normally-distributed noise







**Figure 7.** Eigenvalue of the $\boldsymbol{S}$ matrix, plotted as function of the Eigennumber.

term, $\eta$, added.

$$\eta = \mathcal{N}(0, \sigma^2); \quad \text{where } \sigma = 0.1 \qquad (11)$$

$$\boldsymbol{d}_{\mathrm{nor}} = \boldsymbol{G}\boldsymbol{m}_{\mathrm{nor}} + \eta \qquad (12)$$

$$\boldsymbol{d}_{\mathrm{enh}} = \boldsymbol{G}\boldsymbol{m}_{\mathrm{enh}} + \eta \qquad (13)$$

Recovery of the original model was then attempted using several different techniques. These used standard library functions provided by the Python `numpy` (Eqn. 14) and `scipy` (Eqn. 15) packages (Harris et al., 2020), as well as a direct maximum





a-posteriori estimate (Eqn. 16) and Tikhonov regularised solution (Eqn. 17). For the normal ionosphere case, these are:

$$m_{\mathrm{LS}} = \texttt{linalg.lstsq(G,d)} \tag{14}$$

$$m_{\mathrm{NNLS}} = \texttt{scipy.optimise.nnls(G,d)} \tag{15}$$

$$m_{\mathrm{est}} = (G^T \Sigma^{-1} G)^{-1} G^T \Sigma^{-1} d \tag{16}$$

$$m_{\mathrm{tik}} = V S^{\dagger} U^T d \tag{17}$$

Figure 8 shows the comparison of the original model and the data generated from the inverse solutions. In the figure, the "nominal" data is shown with a solid black line. This is the original electron density profile during normal ionospheric conditions and applying the forward model without noise (Section 2.2). The blue, noisy data is the same, but with the noise term, $\eta$, applied (Equation 12). The dashed traces are for the maximum a-posteriori estimate and Tikhonov-regularised solutions, when transformed with the forward matrix:

$$d_{\mathrm{est}} = G m_{\mathrm{est}} \tag{18}$$

$$d_{\mathrm{tik}} = G m_{\mathrm{tik}} \tag{19}$$

Residuals can be calculated by subtracting the original noiseless data from these inverse problem solutions, for example:

$$r_{\mathrm{est}} = d_{\mathrm{est}} - d \tag{20}$$

These are plotted for the different solution forms (Equations 14–17), as shown in Figure 9. The fact that the residuals are all effectively the same, indicates that all methods are recovering the same solution and that the algorithms are mathematically equivalent.

Note also that the residual values are small, $<0.1\,\mathrm{dB}$, which is considered the noise limit of current instrumentation. The residuals are larger at lower frequencies which is a result of the increased relative importance of the noise with respect to the signal.

Figure 10 shows the maximum a-posteriori together with the original "true" data. The two profiles bear no resemblance. However the $m_{\mathrm{est}}$ does match the same general shape of the collision frequency profile. What is happening is that the collision frequency is a dominant input form, and the solution naturally aligns itself to it. In considering the different methods for





**Figure 8.** Application of the forward models to different solutions. The black 'Nominal' line does not readily appear as the 'Estimate' is colinear with it. The horizontal axis the observing radio frequency, $\nu = \omega/2\pi$.

solving the inverse problem, all of them give repeatable results, based on the input data. The variation between individual solution methods is close to the floating-point quantisation noise. However, even though it satisfies the stability criteria for a well-posed problem, the mathematical examination still implies that it is possible to formulate non-unique solutions.

If the model solution matches the same form as the collision frequency profile (which is an input function) then, in principle, it should be possible to fit any input function to measured absorption.



**Figure 9.** Residuals between data from original and determined models. As all four methods give the same results, the lines are superimposed.

### 3.3 Inversion of arbitrary functions

To demonstrate the ill-posed nature of the inverse problem, a series of arbitrary functions was fitted to simulated electron density profiles. In this case,

$$d = G_s m \tag{21}$$

where $d$ is the simulated data, $G_s$ is the forward matrix for a specific shape function, and $s$ is the shape function parameter. Because the shape function has a fixed altitude profile (e.g. a gaussian can be specified to have a peak at a pre-determined 215 altitude), then the only free parameter is the scaling of the function. Thus, $s$ is effectively a scalar.





**Figure 10.** Fit of an electron density profile (SVD method). The discontinuity corresponds to the minimum altitude of the input collision frequency profile.

The data, $\boldsymbol{d}$, is a function of frequency/polarisation-mode, but the forward model collapses to a single line, as:

$$\boldsymbol{G} = \left[ \frac{k\,\nu_1 f(h_1)\Delta h}{\nu_1^2 + (\omega_1 + \omega_L)^2} + \cdots + \frac{k\,\nu_1 f(h_H)\Delta h}{\nu_1^2 + (\omega_1 + \omega_L)^2} \quad + \quad \frac{k\,\nu_1 f(h_1)\Delta h}{\nu_1^2 + (\omega_1 - \omega_L)^2} + \cdots + \frac{k\,\nu_1 f(h_H)\Delta h}{\nu_1^2 + (\omega_1 - \omega_L)^2} \right] \tag{22}$$

where $f(h)$ is the value of the shape function (where the parameter is the altitude height, $h$). The other terms are as per Equation 6.

As an example of how these operate in practice, two gaussian functions were chosen. The gaussian width parameter in both cases was $\sigma = 5$ and the peak height was set to $\mu = 70$ and $\mu = 100\,\text{km}$ for the low and high cases, respectively. Absorption profiles were created using the physical models indicated in the previous section. Thus a quasi-real atmospheric profile for





both "normal" and "enhanced" ionospheric conditions could be created. For both cases, the arbitrary shape functions were fitted which are shown, together with the results, in Figures 11 and 12, for the low and high gaussian profiles. With the fitted

result, there are also the residuals for the normal and enhanced ionospheric conditions. The residuals have been plotted with the same vertical scaling to allow easy comparison. The range of these scales was set to $\pm 0.1$ dB, corresponding to the approximate riometry noise that could be anticipated from real experimental data (McKay et al., 2015, e.g.)).

In both cases, low- and high-gaussian, the residuals were below the expected noise limit of the riometer. Additionally, there was no significant difference between the two. The implication of this is that the inverse method is incapable of determining

the altitude of a gaussian profile.

Additional testing showed that similar results could be found with other forms (delta functions, gradients, constant offsets, etc.). In all cases, there is insufficient information to be able to say anything meaningful about the height distribution of the electron density profile, given observed absorption over the 18–80 MHz range for o- and x-mode propagation.

### 3.4 Remarks

The original work done on the inversion problem was by Parthasarathy et al. (1963). In that work, three parameters were fitted. Since then, various authors have attempted linear least-squares fitting to determine the maximum likelihood polynomial coefficients (Belikovich et al., 1964, e.g.). However, as has been shown, even in the best-case field-aligned propagation scenario, there is insufficient information to discern a single parameter (gaussian height) fit from the data. The implication is that this line of research is mathematically demonstrated to be unattainable. Nevertheless, there are still other advantages of multi-frequency

riometry measurements. Firstly, it permits more independent measurements of power for the same antenna.

Multi-frequency riometry also validates that detected absorption is the result of electron content in the ionosphere. This is because any absorption from the atmosphere will be a function of frequency ($\omega$), approximating $A(\omega) = A_0/\omega^2$. This makes the instrument robust against natural and artificial forms of radio interference. In the case of natural interference, these may be useful scientific measurements in their own right, such as observation of strong solar radio emissions or Jovian decametric

emissions.

Accurate electron density profile measurements therefore require a strong additional a priori assumption of the electron density profile, such as a parameterised model for the ionisation source or supplementary measurements, such as those from an incoherent scatter radar. The results of this study indicate that the shape of the absorption spectrum does not provide any distinguishable information on the electron density height profile from typical substorm electron precipitation. However,

exceptional cases may still exist where a remarkable part of the ionisation reaches down to below 50 km altitude. From the known ionisation sources, at least the major Solar Proton Events can ionise the atmosphere down to stratospheric altitudes and hence be expected to potentially change the spectral shape (Verronen, 2006). It is also worth mentioning that if trying to distinguish this effect (or any other anomalous ionisation in the deep D-region) from the absorption spectrum, one needs to consider carefully the signal-to-noise ratio (SNR) of the instrument. Although the SNR issue is not the primary focus of this

paper, it is important to mention because of 1) a finite SNR always reduces the absorption detected (the lower the SNR, the lower the absorption measured), 2) the amount of this reduction depends on the magnitude of the absorption itself (the higher

**Figure 11.** Fitting a 70-km peak gaussian electron density profile to simulated normal (nor) and enhance (enh) ionospheric profile data. The residuals are shown in the two top panels. The residuals are shown in the two top panels as a function of the observing frequency: a) normal and b) enhanced, with the o-mode shown in blue and the x-mode in red.





**Figure 12.** Fitting a 100-km peak gaussian electron density profile to simulated normal and enhance ionospheric profile data. The residuals are shown in the two top panels as a function of the observing frequency: a) normal and b) enhanced, with the o-mode shown in blue and the x-mode in red.

the real absorption, the more SNR reduces the detection), and 3) the SNR is frequency dependent for any real spectral riometer instrument. Hence, without taking the SNR into account, the spectral shape apparently changes always when the absorption changes, regardless of the ionisation altitude.

## 4   Conclusions

This study has considered the determination of atmospheric electron density using multi-frequency, multi-polarisation, cosmic noise absorption data (the spectral riometry technique). It has examined the solutions that can be obtained (using both real and modelled data) and has considered if the problem is "well posed".

The determination of electron density profiles, primarily in the D and E region of the ionosphere, can be attempted using
an inverse problem technique and radio absorption data. The absorption is measured over the 15–78 MHz range, which is that which could be achieved with existing instrumentation.

A forward model was developed based on the riometry equation. Using a singular-value decomposition, the electron density profile was solved. However the profile followed the collision-frequency parameter, indicating that it was not well-determined and was thus strongly influenced by other solutions. The assessment of the eigenvalues indicated that there are only a few
significant basis functions, thus no real information could be recovered.

This was further tested by finding maximum-likelihood estimates for arbitrary profile functions. The residuals were of similar form and were contained within the noise range that could be expected from typical riometry data. However, even with zero experimental error, it would not be possible to determine peak profile heights using the frequency and propagation-mode data available. As predicated by the eigenvalue analysis, multiple solutions would exist.

A well-posed inverse problem is one in which:

- a solution to the problem can be found (existence)

- there is only one solution for the problem (uniqueness)

- the solution depends on the data (stability)

In the spectral riometry case, it has been demonstrated that the solutions found are not unique. Therefore the problem is
ill-posed.

Although multi-frequency measurements have other benefits (such as additional independent measurements of power, and thus robustness to radio interference), a typical electron density profile as the result of substorm activity cannot be estimated uniquely from multi-frequency riometer observations alone.

*Author contributions.*   The text and figures were produced by DM. Supervision was provided by JV and NP. The initial collaboration involved
DM, AK and JV. All co-authors contributed references, comments and suggestions for the completion of the work.



*Competing interests.* The authors declare that they have no conflict of interest.

*Disclaimer.* TEXT

*Acknowledgements.* The authors wish to thank A. Kvammen, T. Rexer and B. Gustavsson for their useful discussions. The work by D. McKay is partly supported by the Academy of Finland project number 322535. N. Partamies is supported by the Research Council of Norway under CoE contract 223252 and a research grant contract 287427.




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
