# Peer review of "On the determination of ionospheric electron density profiles using multi-frequency riometry"

_Geoscientific Instrumentation, Methods and Data Systems, 2021_

## Author Comment (AC1)

This response document pertains to the paper *On the determination of ionospheric electron density profiles using multi-frequency riometry*. The authors would like to thank both reviewers for their positive response and constructive criticism. We appreciate the time they have taken to review and help us improve this work. No response was required for Reviewer 1. Included below are Reviewers 2's comments (shown in *green italics*) with the authors' responses interleaved.

— *D. MᶜKay, J. Vierinen, A. Kero, N. Partamies,   25-Nov-2020*
* * *
**Response to Reviewer 2**

Ref: https://doi.org/10.5194/gi-2021-25-RC2

**Reviewer major issues:** "*This study examined derivation of the electron density height profile from measurements with the multi-frequency riometer, and concluded that the profile cannot be estimated uniquely from the measurement alone. I agree with the conclusion and the mathematical process. However, to improve the article, I think that the text had better clearly write unique points different from previous works in this field.*

*Estimation of the electron density profile has been conducted by several researchers in this field for more than half century. They also suggested difficulty in estimating unique profile close to the real one. Then some of the previous works proposed model functions to assume the height profile. Since authors also have worked in this field for many years, I think that you know well, but you can refer to, for example, Parthasarathy et al. (JGR, 1963), Hultqvist (PSS, 1968), Stoker (JGR, 1987), McKay-Bukowski et al. (IEEE Trans. Geosci. Remote Sens., 2015), Cheng et al. (JGR, 2006). Comparing with these previous works, an advantage of this work would be the mathematical approach, which presents difficulty in an objective manner. Section 1 (Introduction) should mention this point.* "

The reviewer is correct that other studies have pointed out the difficulty in estimating unique profiles.

The reviewer suggested five papers. These are:

Parthasarathy et al. (1963) — this paper had already been referenced, but we now mention their conclusion that *"best-fit profiles showed significant differences from theoretical profiles"*.

Hultqvist (1968) —
Stoker (1987) —
Cheng et al. (2006) — These three are all good suggestions, and we now include them to provide the reader with additional leads, should they wish to peruse other works that cover this topic.

McKay-Bukowski et al. (2015) — we do cite this work elsewhere in our paper, but we do not refer to it here in the introduction, as Kero et al. (2014) makes use of the same instrument, but discusses the profile derivation problem in more detail. We feel that is a more succinct and pertinent reference to use at this point in the paper.

As suggested, we now emphasise the point that our work here is a mathematical approach, which explains the difficulties encountered in earlier studies in an objective manner.

**Reviewer comment 2.1:** " *There are some minor comments.*

*L66-68: There are two comments on these lines. (1-1) Can the neutral density change be significant for the collision frequency? It is generally hard to measure the density, so we used to apply the model value, but it is likely different from the true value. (1-2) The electron temperature may significantly increase around 105-110 km with large electric fields or Farley-Buneman instability, although it is not often occurred. So I agree that we can ignore such a special situation, but it may be good to mention in the text. "*

The collision frequency is, by definition, directly proportional to the pressure, so yes a neutral density change can make a difference to a certain degree. Compression of air (pressure change) might also impose a second-order impact via the temperature, but this we can probably neglect.

The follow-up question (implicit to the first one) asks if this bias is capable of making any difference in the absorption. In principle, "yes", but in practise "no".

The CNA is insensitive for any reasonable changes in the collision frequency. For example, doubling the atmospheric pressure at all the altitudes (!) results in a less than 5% error in $A(\text{dB})$ for reasonable electron density levels. Reasonable means here that these $N_e$-levels produce a reasonable $A(\text{dB})$ (like smaller than 30 dB) in the first place.

Another way to look at this question is to realise that the collision frequency is decaying very rapidly in altitude, and typically much faster than $N_e$ or $\nu$ values in the altitudes relevant for the absorption. So, if the pressure is changed up to a factor of 2, the altitude of absorption is merely shifted up/down by a few kilometres, keeping the total absorption more or less constant. We say more or less, because $N_e(h)$ is not always strictly $N_e(h + 5\,\text{kms})$. But the net effect is small, nevertheless.

This reasoning also depends on where the $N_e(h)$ is obtained for calculations in the first place. If it is *modelled* by exposing the atmosphere to some ionising radiation, then the ionisation and the consequent $N_e(h)$ also follow the pressure levels, thus causing a symmetric shift in altitude.

We agree that Farley-Buneman may locally generate strong enhancements in the electron density, which can further affect the collision frequencies and thus the absorption. However, as the referee also points out, the most favourable height range for Farley-Buneman is above about 100km. As absorption is most sensitive to changes below about 100km we consider this as a minor contribution.

In summary, the pressure levels, or temperature changes, or waves or anything like that, have only a marginal effect in CNA. Most importantly, this point doesn't affect by any means to the conclusions made in the paper.

**Reviewer comment 2.2:** " *L86: "where i is the height array index for a given height, h" Redundant. Already written at L85.* "

Fixed.

**Reviewer comment 2.3:** " *L100: Probably "Figure 3".* "

Fixed.

**Reviewer comment 2.4:** " *L119: This may be better to say "why many riometers used to be operated in the 30-40 MHz range".* "

We have corrected this to read *"many riometers"*, as some operate outside of this range. However, we have refrained from adding *"used to be"*, as that implies they are no longer operating, which is not the case.

The text now reads: "The effect of absorption decreases with radio frequency, which is the reason why many riometers operate in the 30–40 MHz range."

**Reviewer comment 2.5:** " *L158: Probably "eleventh" instead of "tenth".* "

Fixed.

**Reviewer comment 2.6:** " *L162-163: "As a result, it can be hypothesized that any number of model solutions could be formulated that would result in a superficial fit of the data." Does this suggest that a model reproduced by some lower terms may be acceptable to fit the data?* "

As *any* profile could be used, any model using only lower terms does not not constrain the function shape. And, reducing the formulation to only the lowest term offers no information about the shape, only the magnitude of the integrated effect. This is what a single-frequency riometer provides with a single absorption measurement. This statement has been added to the text.

**Reviewer comment 2.7:** " *No solid black line in Figure 8.* "

There are solid black lines, but they are masked by the other lines. We had tried to indicate this in the caption. We have now changed the layers to make the black line more apparent. However, the black line is effectively co-linear with the red dashed line.

**Reviewer comment 2.8:** " *L195: (8-1) There are two curves in Figure 9. The text should mention each line. (8-2) My understanding is that the absorption residual, shown in Fig. 9, was derived from Fig. 8 as differences between values with solid and dashed lines. A simple calculation may provide noisy results but curves in Fig. 9 are very smooth. Why?* "

The two lines are the o-mode and x-mode. The caption and text have both been amended to make this clear. The plot has also been redone (in order to make the x-mode line dashed). A slightly different form appears (due to the difference in the noise), however the main point is that the lines for the four methods give exactly the same results for the two modes. The reasons why these lines are smooth is that they are for the models (not the noise).

**Reviewer comment 2.9:** " *L200 "The two profiles bear no resemblance." The SVD may not give unique solution. Is there any possibility that there is/are other answer(s)? It might be closer to the true data. If this is the case, you do not need to pick up the m_est curve as the representative one.* "

The solution is stable, but the collision frequency is a dominant input, so the solution locks to that. We mention this, as well as the fact that all of methods give repeatable results. However, we now add the word *"stable"*, to help make this clearer.

**Reviewer comment 2.10:** " *L213 "s is the shape function parameter." I cannot find "s" in the equation (21). Since the last sentence of this paragraph says "s" is a scalar, do you assume s = 1 so invisible in the equation?* "

The use of $\vec{G}_s$ is to indicate that the forward matrix in this case is the forward matrix for a specific shape function $\vec{s}$. We have propagated this nomenclature to the subsequent equations and have added a clarification to the text.

**Reviewer comment 2.11:** " *Eq.(22): I cannot follow derivation of the equation 22, in particular, about the relation with the equation 21. What relations are there between $G_s$ and $G$?* "

This has been fixed by using consistent symbols between Eqns. 21 and 22. In particular, we have replaced the arbitrary $f()$ function with the shape function $s()$ and have indicated that the $\vec{G}_s$ in Eqn. 22 is indeed the same as the the one in Eqn. 21. See also our response to the previous comment.

**Reviewer comment 2.12:** " *Caption of Figure 11: "The residuals are shown in the two top panels."* *Redundant.*"

Fixed.

**Reviewer comment 2.13:** " *L265: "which is that" Should remove.* "

This was what we had intended, but we accept that it may appear ambiguous or awkward. We have thus reformulated the sentence to make it clearer.

$* * *$

**References**

Cheng, Z., Cummer, S. A., Baker, D. N., and Kanekal, S. G.: Nighttime D region electron density profiles and variabilities inferred from broadband measurements using VLF radio emissions from lightning, Journal of Geophysical Research: Space Physics, 111, https://doi.org/https://doi.org/10.1029/2005JA011308, URL `https://agupubs.onlinelibrary.wiley.com/doi/abs/10.1029/2005JA011308`, 2006.

Hultqvist, B.: On the solution of the integral equation relating height distribution of electron density to radio-wave absorption, Planetary and Space Science, 16, 529–537, https://doi.org/10.1016/0032-0633(68)90095-0, 1968.

Kero, A., Vierinen, J., McKay-Bukowski, D., Enell, C.-F., Sinor, M., Roininen, L., and Ogawa, Y.: Ionospheric electron density profiles inverted from a spectral riometer measurement, Geophys. Res. Lett., 41, 5370–5375, https://doi.org/10.1002/2014GL060986, 2014.

McKay-Bukowski, D., Vierinen, J., Virtanen, I. I., Fallows, R., Postila, M., Ulich, T., Wucknitz, O., Brentjens, M., Ebbendorf, N., Enell, C., Gerbers, M., Grit, T., Gruppen, P., Kero, A., Iinatti, T., Lehtinen, M., Meulman, H., Norden, M., Orispää, M., Raita, T., de Reijer, J. P., Roininen, L., Schoenmakers, A., Stuurwold, K., and Turunen, E.: KAIRA: The Kilpisjärvi Atmospheric Imaging Receiver Array — System Overview and First Results, Geoscience and Remote Sensing, IEEE Transactions on, 53, 1440–1451, https://doi.org/10.1109/TGRS.2014.2342252, 2015.

Parthasarathy, R., Lerfald, G. M., and Little, C. G.: Derivation of Electron-Density Profiles in the Lower Ionosphere Using Radio Absorption Measurements at Multiple Frequencies, J. Geophys. Res., 68, 3581–3588, https://doi.org/10.1029/JZ068i012p03581, 1963.

Stoker, P. H.: Riometer absorption and spectral index for precipitating electrons with exponential spectra, Journal of Geophysical Research, 92, 5961–5968, https://doi.org/10.1029/JA092iA06p05961, 1987.